# BiC-Occ: Bi-directional Circulated 3D Occupancy Prediction for Autonomous Driving

## Abstract

Vision-based 3D occupancy prediction is the cornerstone in autonomous driving systems to provide comprehensive scene perception for subsequent decisions, which requires assessing voxelized 3D scenes with multi-view 2D images. Existing methods mainly adopt unidirectional pipelines projecting image features to BEV representations for following supervision, whose performances are limited by the sparsity and ambiguity of voxel labels. To address this issue, we propose a **Bi**-directional **C**irculated 3D Occupancy Prediction (**BiC-Occ**) framework for more accurate voxel predictions and supervisions. Specifically, we design a Bi-directional View Transformer module that approximates invertible transition matrices of the view transformation process, promoting the self-consistency between 2D image features and 3D BEV representations. Furthermore, we propose a Circulated Interpolation Predictor module that exploits local geometric structures to align multi-scale BEV representations, correcting local ambiguity with consistent occupancy predictions across different resolutions. With the synergy of these two modules, the self-consistency within different perception views and occupancy resolutions compensates for the sparsity and ambiguity of voxel labels, leading to more accurate 3D occupancy predictions. Extensive experiments and analyses demonstrate the effectiveness of our BiC-Occ framework.

## 1 Introduction

Perceiving the 3D geometry of the surrounding scene accurately serves as a fundamental ability for autonomous driving systems. Although the LIDAR sensor can directly capture geometry-aware data with precise depth information, it suffers from high implementation costs and sparse scanned points, which restricts its further development. Recently, vision-based 3D scene perception has been emerging as a promising alternative to LIDAR-based one due to its cost-effectiveness. Taking multi-camera images as input, the main challenge of vision-based 3D scene perception is to transform 2D images into 3D scenes.

To compensate for the lack of depth information in the input images, conventional voxel-based methods Zhou & Tuzel (2018); Zhu et al. (2021) divide the 3D space into discrete voxels and assign a feature vector to each voxel as its representation. Voxel-based methods have achieved great performance in LIDAR-based 3D scene perception tasks such as lidar segmentation Liong et al. (2020); Cheng et al. (2021); Ye et al. (2023) and 3D scene completion Cao & de Charette (2022); Chen et al. (2020); Yan et al. (2021); Li et al. (2023b). Recently, Monoscene Cao & de Charette (2022) first generalizes voxel-based methods to 3D scene reconstruction with only RGB inputs, and TPV-Former Huang et al. (2023) further extends to the 3D occupancy prediction task with multi-camera inputs. However, voxel-based methods need to take each single voxel into consideration, which leads to a high computation burden, limiting its performance in larger scenes.

Towards a more computationally efficient pipeline for 3D scene perception, the BEV-based methods have attracted more attention from researchers. Considering that the height dimension contains less information than the other two dimensions in 3D scene representations, BEV-based methods compress height dimension into each BEV grid to generate more compact representations capturing height information implicitly Lang et al. (2019). To complete 2D input images with depth-wise information, recent research on BEV-based methods can be mainly classified into two kinds, regarding whether the depth information is computed implicitly or explicitly. BEVFormer Li et al. (2022)

is a representative work that learns depth information implicitly with pre-defined grid-shaped BEV queries. The other line of works mainly follows the Lift-Splat-Shoot (LSS) Philion & Fidler (2020) paradigm to explicitly generate depth estimation for input images Huang et al. (2021); Reading et al. (2021); Zhang et al. (2022); Liu et al. (2023). Efforts have been made to improve depth estimation with direct depth loss supervision Li et al. (2023d) and dynamic temporal stereo information Li et al. (2023c).

However, the aforementioned methods mostly adopt unidirectional pipelines supervised by annotated ground truth, which suffers from the sparsity and ambiguity of voxel labels. (1) The sparsity of voxel labels stems from the characteristic that a large portion of voxels are empty in real-world scenarios, which fails to provide comprehensive supervision for the view transformation process. (2) The ambiguity of voxel labels roots in the inevitable errors from manual annotations and resolution downsampling, which limits the final occupancy prediction performance. To address the above issues, we propose a *Bi-directional Circulated 3D Occupancy Prediction* (**BiC-Occ**) framework, which aims at promoting the self-consistency within different perception views and occupancy resolutions to alleviate the sparsity and ambiguity of voxel labels. First, we introduce the *Bi-directional View Transformer* (**Bi-VT**) to address the sparsity of voxel labels through constructing reversible and self-consistent view transformations. The procedure begins with a Forward Mapping block and a Backward Sampling block modeling the 2D-to-3D mapping and 3D-to-2D sampling distributions respectively. Then, the Invertible Refinement block further approximates invertible transition matrices through tensor decomposition and recovery, leading to reversible view transformations with self-consistency. Second, we present the *Circulated Interpolation Predictor* (**CIP**) to address the ambiguity of voxel labels by promoting the alignment among multi-scale BEV representations. Specifically, the module starts with a Geometric Interpolation block aligning multi-scale voxel representations concerning local geometric structures. Then, we design a Circulated Loss to promote the consistency among multi-scale voxel representations, thereby generating more accurate 3D occupancy predictions of different voxel grid resolutions and mitigating the ambiguity of voxel labels. Extensive experiments and analyses validate the effectiveness of our proposed BiC-Occ framework.

The main contributions are summarized as follows:

- We identify the inherent sparsity and ambiguity challenges of voxel labels in 3D occupancy prediction, and propose the BiC-Occ approach to address them.

- The Bi-directional View Transformer module addresses the sparsity of voxel labels through learning invertible transition matrices via tensor decomposition and recovery for reversible view transformations with self-consistency.

- The Circulated Interpolation Predictor module addresses the ambiguity of voxel labels through alignment among multi-scale voxel representations, coupling with a Circulated Loss for more accurate 3D occupancy predictions of different occupancy resolutions.

## 2 PROBLEM FORMULATION

The objective of 3D occupancy prediction is to assess the voxelized 3D occupancy $O$ of surrounding scenes given multiple surround-view image inputs $\{I_i\}_{i=1}^{N_c}$, where $N_c$ denotes the number of cameras. Existing occupancy prediction frameworks mainly consist of three components: Image Encoder, View Transformer, and Occupancy Predictor. We formulate their functions as follows:

### 2.1 IMAGE ENCODER

The Image Encoder usually consists of a pretrained image backbone (*e.g.*, ResNet-50 He et al. (2016)) and a feature pyramid network for extracting the surround-view 2D image features $F_{\text{img}} \in \mathbb{R}^{N_c \times C \times H \times W}$, where $C$ denotes the embedding dimensions of the feature space, and $(H, W)$ represents the scale of 2D feature maps.

### 2.2 VIEW TRANSFORMER

The View Transformer is a fundamental module in occupancy frameworks that transforms 2D image features $F_{\text{img}}$ to 3D BEV representations $F_{\text{BEV}} \in \mathbb{R}^{C \times X \times Y \times Z}$, where $(X, Y, Z)$ denotes the target

resolution of 3D volumes. There are two main patterns: explicit view transformation (EVT) and implicit view transformation (IVT). EVT methods Philion & Fidler (2020); Huang et al. (2021) first calculate explicit depth distribution maps $D_{\text{img}}$ of 2D image features, then conduct voxel pooling on the outer product $F_{\text{img}} \otimes D_{\text{img}}$ to generate 3D BEV representations. On the other hand, IVT methods Li et al. (2022); Wang et al. (2022) directly learn implicit mapping relationships between the 2D feature maps and 3D voxel grids with BEV queries and corresponding sampling offsets. To promote reversible view transformations, we propose the following assumption and proposition for general formulations and theoretical insights.

**Assumption 1.** *Let $A_{\text{VT}} \in \mathbb{R}^{HW \times XYZ}$ denote the general transition matrix for view transformation, i.e., $F_{\text{BEV}} = F_{\text{img}} \cdot A_{\text{VT}}$, for both EVT and IVT methods, we can factorize the transition matrix as the Kronecker product of two transition score matrices and formulate the view transformation process as follows:*

$$F_{\text{BEV}} = F_{\text{img}} \cdot A_{\text{VT}} = F_{\text{img}} \cdot (A_{\text{img}} \otimes A_{\text{BEV}}) \tag{1}$$

*where $A_{\text{img}} \in \mathbb{R}^{H \times W}, A_{\text{BEV}} \in \mathbb{R}^{X \times Y \times Z}$ denote the 2D and 3D transition score matrices respectively, and $\otimes$ represent the Kronecker product operation.*

The insight behind the assumption is that the essence of view transformation is to learn the correspondence among 2D pixels and 3D voxels, which can be considered as calculating the similarity score regarding each 2D pixel and 3D voxel. Therefore, we further decompose the procedure as first generating score matrices of 2D image features and 3D BEV representations respectively, then calculating the transition matrix with Kronecker product for pixel-voxel similarity scores.

**Proposition 1.** *Under previous Assumption 1, a reversible view transformation requires an invertible transition matrix, which is equivalent to invertible 2D and 3D transition score matrices. The reverse view transformation can be formulated as follows:*

$$F_{\text{img}} = F_{\text{BEV}} \cdot A_{\text{VT}}^{-1} = F_{\text{BEV}} \cdot (A_{\text{img}}^{-1} \otimes A_{\text{BEV}}^{-1}) \tag{2}$$

**Proof.** This follows directly from the property of Kronecker product, that $A \otimes B$ is invertible if and only if $A$ and $B$ are invertible, and the inverse is given by $(A \otimes B)^{-1} = A^{-1} \otimes B^{-1}$. $\qquad \square$

### 2.3 OCCUPANCY PREDICTOR

The Occupancy Predictor takes the BEV representations $F_{\text{BEV}}$ as input and generates the 3D occupancy prediction results $O \in \mathbb{R}^{N_{\text{cls}} \times X \times Y \times Z}$, where $N_{\text{cls}}$ denotes the number of candidate classes, the value of $N_{\text{cls}}$ is set to 2 for the scene completion (SC) task and 17 for the semantic scene completion (SSC) task.

## 3 APPROACH

Figure 1 illustrates the proposed Bi-directional Circulated 3D Occupancy Prediction (BiC-Occ) framework, which consists of three key components: (1) an Image Encoder for extracting 2D image features, (2) a Bi-directional View Transformer (Bi-VT) module that addresses the sparsity of voxel labels by approximating an invertible transition matrix through tensor factorization and recovery for reversible view transformation with self-consistency, (3) a Circulated Interpolation Predictor (CIP) module that addresses the ambiguity of voxel labels via leveraging local geometric structures to align different occupancy resolutions.

### 3.1 BI-DIRECTIONAL VIEW TRANSFORMER

The Bi-directional View Transformer (Bi-VT) module consists of three blocks to approximate an invertible transition matrix for addressing the sparsity of voxel labels. The Forward Projection and Backward Projection blocks first generate bi-directional 2D-to-3D mapping and 3D-to-2D sampling distributions respectively, extracting transition score matrices for the following tensor factorization and recover. Then the Invertible Refinement block adopts vector-matrix decomposition and truncated singular value decomposition to factorize and recovery the principal parts of the forward and backward projection to approximate reversible view transformation.

Figure 1: The overall architecture of our BiC-Occ framework. The Bi-directional View Transformer (Bi-VT) module approximates the invertible transition matrix through tensor factorization and recovery. The Circulated Interpolation Predictor (CIP) module leverages local geometric structures to align different occupancy resolutions for alleviating ambiguity in occupancy prediction results.

**Forward Projection.** To model the forward 2D-to-3D mapping process, we follow the explicit view transformation pipelines Philion & Fidler (2020); Huang et al. (2021), where the 2D pixels take the initiative in view transformation and the 3D voxels passively accept features from the images. Specifically, given the extracted image features $F_{\text{img}}$ and depth distribution maps $D_{\text{img}}$, we utilize fully connected (FC) layers to distill the feature and depth vectors at each coordinate into a single score value:

$$S_{\text{feat}} = \text{FC}(F_{\text{img}}), \quad S_{\text{depth}} = \text{FC}(D_{\text{img}}) \tag{3}$$

where $S_{\text{feat}}, S_{\text{depth}}$ denote the feature and depth score maps respectively, indicating the significance of each coordinate with respect to the feature space and depth dimension. Then, we compute the 2D and 3D transition score matrices as follows:

$$A_{\text{img}}^{\text{fore}} = S_{\text{feat}} \cdot S_{\text{depth}}, \quad A_{\text{BEV}}^{\text{fore}} = \text{GAP}(F_{\text{BEV}}) \tag{4}$$

where $\text{GAP}(\cdot)$ denotes the global average pooling layer for distilling the transition scores at each voxel grid.

**Backward Projection.** To calculate the backward 3D-to-2D sampling functions, we adopt the implicit view transformation frameworks Li et al. (2022); Wang et al. (2022), where the 3D voxels are filled with initial query values and then project 3D points back onto the images with sampling offsets. Specifically, the 2D and 3D transition matrices are computed with 2D and 3D global average pooling layers as follows:

$$A_{\text{img}}^{\text{back}} = \text{GAP}(F_{\text{img}}), \quad A_{\text{BEV}}^{\text{back}} = \text{GAP}(F_{\text{BEV}}) \tag{5}$$

**Invertible Refinement.** An ideal view transformation pipeline is to generate a reversible projection from 2D image features to 3D voxel representations. However, the high rank of the transition matrix and the sparsity of voxel labels hinders efficient optimization and accurate supervision for learning reversible view transformations. To approximate reversible view transformations and invertible transition matrices and improve the efficiency and accuracy of supervisions, we first adopt vector-matrix (VM) decomposition to lower the dimension of 3D transition score matrices, then we utilize the truncated singular value decomposition (T-SVD) further approaching invertible matrices. Specifically, considering that the height dimension provides less information compared to the other two dimensions, we decompose the 3D voxel space along the vertical axis and horizontal plane:

$$A_{\text{BEV}}^{\text{VT}} = \sum_i A_{Zi}^{\text{VT}} \circ A_{XYi}^{\text{VT}} \tag{6}$$

where $\text{VT} \in \{\text{fore}, \text{back}\}$ denotes the type of 3D transition score matrices, $A_{Zi}^{\text{VT}}$ represents the vertical factor, and $A_{XYi}^{\text{VT}}$ is the horizontal factor. Then we conduct the truncated singular value

decomposition on the horizontal factor, where the top-$k$ singular values and corresponding eigenvectors are selected for the recovery of matrices:

$$U_i^{\text{VT}}, \Sigma_i^{\text{VT}}, V_i^{\text{VT}} = \text{T} - \text{SVD}(A_{XYi}^{\text{VT}}|k) \tag{7}$$

where $k$ is the truncated thresholds, $\Sigma_i^{\text{VT}}$ denotes the diagonal matrix of the top-$k$ singular values, and $U_i^{\text{VT}}, V_i^{\text{VT}}$ represent the matrices of left and right eigenvectors respectively. Finally, our approximation of the invertible transition matrix is recovered as follows:

$$A_{inv} = \sum_{\text{VT}} A_{\text{img}}^{\text{VT}} \otimes \sum_i A_{Zi}^{\text{VT}} \circ (U_i^{\text{VT}} \Sigma_i^{\text{VT}} V_i^{\text{VT}}) \tag{8}$$

Thus, we are able to conduct approximately reversible view transformations as follows:

$$F_{\text{BEV}} = F_{\text{img}} \cdot A_{inv} \tag{9}$$

which addresses the sparsity of voxel labels with VM decomposition reducing the matrix rank and T-SVD improving information density, enabling more efficient and accurate supervision.

## 3.2 CIRCULATED INTERPOLATION PREDICTOR

The Circulated Interpolation Predictor (CIP) module is proposed to address the ambiguity of voxel labels by aligning multi-scale BEV representations. The Geometric Interpolation block is first adopted to align multi-scale BEV representations regarding local geometric structures in a circulated manner. Then we design the Circulated Loss as supervision of both geometric similarity and prediction accuracy among different occupancy resolutions, correcting the ambiguous voxels with consistency across different occupancy resolutions.

**Geometric Interpolation.** The Geometric Interpolation block aims to address the ambiguity of voxel labels by leveraging local geometric structures. For instance, within a $3 \times 3 \times 3$ voxel cube, if all 26 surrounding voxels are classified as "vegetation", it is highly probable that the central voxel also belongs to the "vegetation" class, irrespective of its initial predicted occupancy.

Specifically, suppose we are given two BEV representations with different resolutions, termed as $F_{\text{BEV}}^h \in \mathbb{R}^{C \times X^h \times Y^h \times Z^h}$ with higher resolution and $F_{\text{BEV}}^l \in \mathbb{R}^{C \times X^l \times Y^l \times Z^l}$ with lower resolution. The geometric interpolation block works in a circulated manner, conducting both down-scale and up-scale alignment. (1) *Down-scale alignment* gathers high-resolution voxels in a cubic area as a single low-resolution voxel. To generate more accurate low-resolution voxel semantics, we first adopt the 3D convolution layer to compute the geometric gathering score (Geo-Gather Score) $G_{\text{gather}}$ as abstract representations of geometric structures within each cubic area of high-resolution voxels. Then, we apply the downsample layer with average pooling to get initial downsample result, whose product with $G_{\text{gather}}$ is computed as the result of down-scale alignment. The above process can be formulated as follows:

$$G_{\text{gather}} = 3\text{DConv}(F_{\text{BEV}}^h), \quad F_{\text{BEV}}^{\text{down}} = F_{\text{BEV}}^l + \alpha \cdot G_{\text{gather}} \cdot \text{Down}(F_{\text{BEV}}^h) \tag{10}$$

where $3\text{DConv}(\cdot)$ denotes the 3D convolution layer, $\text{Down}(\cdot)$ represents the downsample layer, and $F_{\text{BEV}}^{\text{dwon}}$ is the down-scale alignment output. (2) *Up-scale alignment* scatters a single low-resolution voxel into a cubic area of high resolution voxels. To generate more reasonable local geometric structures within scattered cubics, we first utilize the transpose 3D convolution layer to calculate the geometric scattering score (Geo-Scatter Score) $G_{\text{scatter}}$, modeling the correlations among the source voxel and scattered cubic voxels. Then, we adopt the upsample layer with trilinear interpolations to generate initial upsample representations, whose product with $G_{\text{scatter}}$ is computed as the up-scale alignment output. The above procedure is formulated as follows:

$$G_{\text{scatter}} = \text{T} - 3\text{DConv}(F_{\text{BEV}}^l), \quad F_{\text{BEV}}^{\text{up}} = F_{\text{BEV}}^h + \alpha \cdot G_{\text{scatter}} \cdot \text{Up}(F_{\text{BEV}}^l) \tag{11}$$

where $\text{T} - 3\text{DConv}(\cdot)$ denotes the transpose 3D convolution layer, $\text{Up}(\cdot)$ represents the upsample layer, and $F_{\text{BEV}}^{\text{up}}$ is the up-scale alignment output, $\alpha$ is the shared weight hyper-parameter.

**Circulated Loss.** To cope with the Circulated Interpolation block, we further design the Circulated Loss as the supervision of both prediction accuracy and geometric similarity among different occupancy resolutions:

$$L_{\text{Circ}} = L_{\text{CE}}(F_{\text{BEV}}^{up}, V^h) + L_{\text{CE}}(F_{\text{BEV}}^{down}, V^l) + \beta \cdot L\text{sim}(F_{\text{BEV}}^{up}, F_{\text{BEV}}^{down}) \tag{12}$$

where $L_{\text{sim}}(\cdot, \cdot)$ represents the similarity loss function, $L_{\text{CE}}(\cdot, \cdot)$ denotes the cross entropy loss function, and $V^h, V^l$ is the stand for the voxel labels of higher and lower resolutions respectively, $\beta$ is the weight hyper-parameter. The cross-entropy loss provides direct prediction accuracy supervision for general optimization on occupancy predictions with different resolutions. On the other hand, similarity loss is adopted to correct local ambiguity by promoting self-consistency among the local geometric structures of different resolutions.

## 4 EXPERIMENTS

In accordance with existing 3D occupancy prediction methods, extensive experiments and analyses are conducted to validate the BiC-Occ framework on the Occ3d-nuScenes dataset Tian et al. (2024). The subsequent sections provide details on the experimental setup, result comparisons, and corresponding analyses.

### 4.1 EXPERIMENTAL SETUP

**Dataset.** Occ3d-nuScenes Tian et al. (2024) is a large-scale autonomous dataset, which provides validation occupancy ground truth labels as a supplement to the popular nuScenes dataset Caesar et al. (2020). The dataset includes 700 scenes for training and 150 scenes for validation, where each frame contains six surround-view RGB images with voxel-wise semantic occupancy labels. The occupancy supervision scope ranges in $[-40m, 40m]$ for the $X, Y$ axis and $[-1m, 5.4m]$ for the $Z$ axis. The original surround-view images are with size $900 \times 1600$, which we resized to the size of $254 \times 704$ as input. The output occupancy predictions are in $200 \times 200 \times 16$ shape with a voxel size of $0.4m$.

**Evaluation Metrics.** Following the evaluation metric in Tian et al. (2024), we adopt the standard IoU metric, ignoring the semantic classes of occupied voxels, for the scene completion (SC) task and the mIoU metric over all semantic classes for the semantic scene completion (SSC) task.

$$\text{IoU} = \frac{TP}{TP + FP + FN}, \quad \text{mIoU} = \frac{1}{C} \sum_{c=1}^{C} \frac{TP_c}{TP_c + FP_c + FN_c} \tag{13}$$

where $TP, FP, FN$ represent the number of true positive, false positive, and false negative occupancy predictions, and $C$ stands for the total number of classes.

**Implementation Details.** For all experimental settings, our BiC-Occ framework is trained with a batch size of 8 on 4 NVIDIA A6000 GPUs, and adopts AdamW Loshchilov & Hutter (2017) optimizer with a learning rate of $2 \times 10^{-4}$ and a weight decay of $0.01$. To be consistent with existing methods Tian et al. (2024); Huang & Huang (2022), we adopt ResNet-50 He et al. (2016) as image backbones, where the input images are resized to $256 \times 704$. Following Huang et al. (2021), we adopt image augmentations as well as BEV data augmentations including random scaling, random cropping, random rotation, and random flipping. We train our models for 24 epochs before evaluating them for the 3D occupancy prediction task.

### 4.2 EXPERIMENTAL RESULTS

Table 1 presents the 3D occupancy prediction results on the Occ3d-nuScenes validation dataset, where our BiC-Occ approach achieves the state-of-the-art performance with $0.5\%$ improvement in Intersection over Union (IoU) for the scene completion (SC) task and $0.1\%$ increase in mean Intersection over Union (mIoU) for the semantic scene completion (SSC) task. The performance improvements of our BiC-Occ approach are attributed to the mitigation of sparsity and ambiguity of voxel labels. Specifically, the Bi-VT module addresses the sparsity of voxel labels with tensor factorization and recovery for reversible view transformation with self-consistency between 2D

Table 1: **3D occupancy prediction results on the Occ3d-nuScenes validation dataset.** Best results are highlighted in bold, and the second-best results are underlined.

| Method | Venue | Image Backbone | Image Size | Epoch | IoU (%) | mIoU (%) |
|---|---|---|---|---|---|---|
| BEVFormer Li et al. | ECCV'22 | ResNet-101 | 928×600 | 24 | - | 26.9 |
| CTF-Occ Tian et al. | arXiv'23 | ResNet-101 | 928×600 | 24 | - | 28.5 |
| TPVFormer Huang et al. | CVPR'23 | ResNet-50 | 900×1600 | 24 | 66.8 | 34.2 |
| SurroundOcc Wei et al. | ICCV'23 | ResNet-101 | 900×1600 | 24 | 65.5 | 34.6 |
| OccFormer Zhang et al. | ICCV'23 | ResNet-50 | 256×704 | 24 | 70.1 | 37.4 |
| BEVDet4D Huang & Huang | arXiv'22 | ResNet-50 | 384×704 | 24 | 73.8 | 39.3 |
| VoxFormer Li et al. | CVPR'23 | ResNet-101 | 900×1600 | 24 | - | 40.7 |
| FBOcc Li et al. | ICCV'23 | ResNet-50 | 256×704 | 20 | - | 42.1 |
| COTR Ma et al. | CVPR'24 | ResNet-50 | 254×704 | 24 | 75.0 | 44.5 |
| **BiC-Occ** | **ours** | ResNet-50 | 254×704 | 24 | **75.5** | **44.6** |

image features and 3D BEV representations. Additionally, the CIP module resolves the ambiguity of occupancy predictions with a circulated alignment across multi-scale BEV representations, promoting consistency across different occupancy resolutions for the correction of local ambiguity. Together, these complementary modules address the sparsity and ambiguity of voxel labels for more accurate 3D occupancy prediction. COTR Ma et al. (2024) integrates the above two patterns into a Geometry-aware Occupancy Encoder, generating compact occupancy representations for better performance.

Table 2: Ablation study on the Occ3d-nuScenes dataset of different components of our BiC-Occ.

| Method | SC IoU | SSC mIoU | barrier | bicycle | bus | car | const. veh. | motorcycle | pedestrian | traffic cone | trailer | truck | drive. suf. | other flat | sidewalk | terrian | manmade | vegetation |
|---|---|---|---|---|---|---|---|---|---|---|---|---|---|---|---|---|---|---|
| Baseline | 71.21 | 39.58 | 46.38 | 26.74 | 44.86 | 51.72 | 26.02 | 27.09 | 27.6 | 29.04 | 31.92 | 38.47 | 80.69 | 40.46 | 51.2 | 54.11 | 45.66 | 39.96 |
| Bi-VT | 74.75 | 43.24 | 50.2 | 31.39 | 45.99 | 54.29 | 30.37 | 31.57 | 29.74 | 33.8 | 35.34 | 41.05 | 83.66 | 45.58 | 55.29 | 58.74 | 50.59 | 45.0 |
| CIP | 74.38 | 43.51 | 51.03 | 31.25 | 45.32 | 54.91 | 29.71 | 32.28 | 29.98 | 34.13 | 36.61 | 42.04 | 83.74 | 46.35 | 55.9 | 58.18 | 50.35 | 44.98 |
| BiC-Occ | **75.5** | **44.6** | **52.23** | **32.73** | **46.38** | **55.72** | **30.6** | **32.98** | **30.7** | **35.76** | **37.6** | **43.12** | **84.21** | **47.12** | **56.63** | **59.76** | **52.23** | **46.45** |

## 4.3 ABLATION STUDY

To validate the contributions of different components of our proposed BiC-Occ approach, we conduct ablation experiments on the Occ3d-nuScenes validation dataset. We gradually integrate the Bi-directional View Transformer (Bi-VT) module and the Circulated Interpolation Predictor (CIP) module into the baseline method Huang & Huang (2022), and the results are illustrated in Table 2. It can be observed that adding Bi-VT enhances the 3D occupancy prediction performance by $3.54\%$ in IoU and $3.66\%$ in mIoU. Incorporating CIP further yields performance improvements of $3.17\%$ IoU and $3.93\%$ mIoU over the baseline. These results demonstrate the effectiveness of promoting self-consistency within different perception views and occupancy resolutions for addressing the sparsity and ambiguity of voxel labels. Furthermore, the Bi-VT module and CIP module show synergistic effects, together leading to superior performance with $4.29\%$ IoU and $5.02\%$ mIoU improvement over the baseline method.

## 4.4 PARAMETER ANALYSES

To further investigate the effectiveness of our BiC-Occ approach, we conduct parameter analyses of the weight hyper-parameter $alpha$ and $beta$ for the Geometric Interpolation block and Circulated Loss respectively. Table 3 presents the experimental results with various values of $\alpha$. Setting $\alpha$ to 0 equals the traditional interpolations without geometric structure information, suffering from local ambiguity. However, with positive $\alpha$ values, local geometric structures are incorporated for better alignment across different occupancy resolutions, correcting local ambiguity for improved performance. We evaluate the impact of $\beta$ for the Circulated Loss in table 4. It can be observed that the similarity loss term improves the occupancy performance by constraining the geometric

Table 3: Parameter analyses on the Occ3d-nuScenes dataset examining the impact of weight hyper-parameter $\alpha$.

| $\alpha$ | IoU(%) | mIoU(%) |
|---|---|---|
| 0 | 75.1 | 43.8 |
| 0.3 | 75.2 | 44.0 |
| 0.5 | 75.3 | 44.2 |
| 1.0 | **75.5** | **44.6** |

Table 4: Parameter analyses on the Occ3d-nuScenes dataset examining the impact of weight hyper-parameter $\beta$.

| $\beta$ | IoU(%) | mIoU(%) |
|---|---|---|
| 0 | 74.9 | 43.9 |
| 0.3 | 75.3 | 44.2 |
| 0.5 | **75.5** | **44.6** |
| 1.0 | 75.2 | 44.3 |

consistency within different occupancy resolutions. For optimal performance, we set $\alpha = 1.0$ and $\beta = 0.5$ in our BiC-Occ framework.

## 4.5 VISUALIZATIONS

Figure 2 demonstrates the visualization results from the Occ3d-nuScenes validation dataset. The surround-view input images are illustrated in the first and third lines. In the first row, the occupancy ground truth is outlined with blue boxes. The second row presents the occupancy predictions generated by the baseline method, where false predictions are indicated with black boxes. While the third row displays the results of our BiC-Occ approach, and orange boxes highlight our refinement for more accurate occupancy predictions. The above qualitative analyses validate the effectiveness of our BiC-Occ framework for improving 3D occupancy prediction performance.

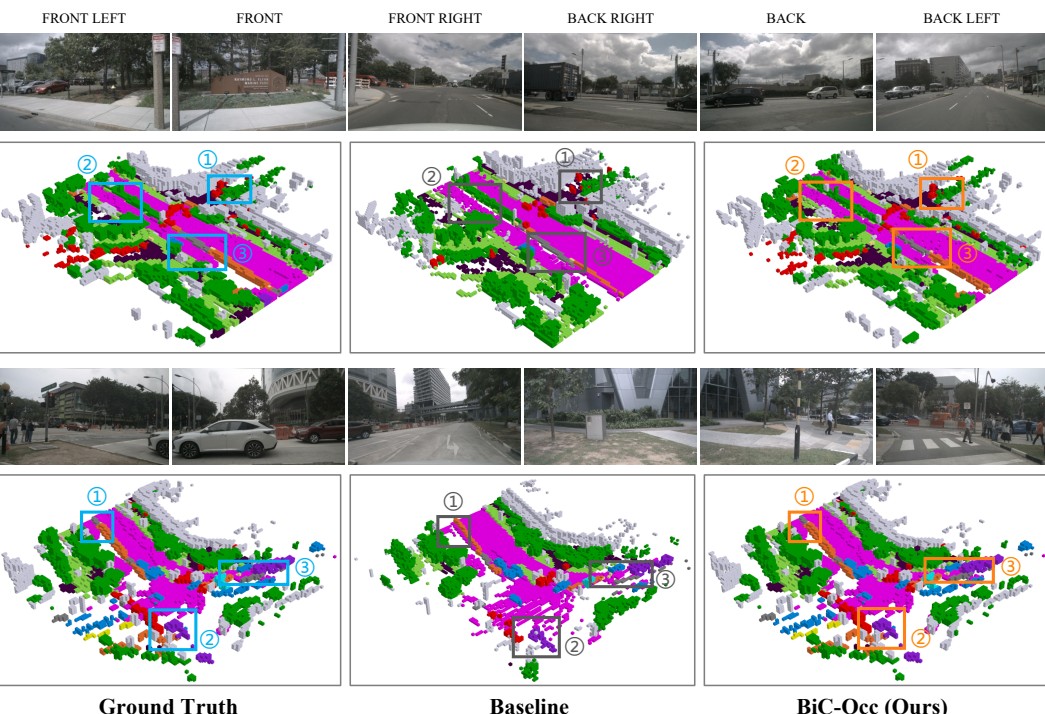

Figure 2: Visualization results on the Occ3d-nuScenes validation dataset. The occupancy ground truth is outlined with blue boxes. While black boxes indicate erroneous occupancy predictions of the baseline method, and orange boxes highlight more accurate predictions by our BiC-Occ. Better viewed when zoomed in.

## 5 RELATED WORK

In this section, we briefly review the literature on two aspects related to this paper: voxel-based scene representation and BEV-based scene representation. Voxel-based methods are popular in LIDAR-based scene perception, while BEV-based methods have attracted more attention in vision-based scene perception due to their computation efficiency.

### 5.1 VOXEL-BASED SCENE REPRESENTATION

Obtaining an effective representation of a 3D scene is a pivotal procedure in the field of autonomous driving. One prominent pattern is voxel-based scene representation, which discretizes the 3D space into voxels and assigns a feature vector to represent each voxel Zhou & Tuzel (2018); Zhu et al. (2021). This technique excels in constructing fine-grained 3D scene structures, and has empowered the success of several tasks such as lidar segmentation Liong et al. (2020); Tang et al. (2020); Cheng et al. (2021); Ye et al. (2021; 2023) and 3D scene completion Cao & de Charette (2022); Roldao et al. (2020); Chen et al. (2020); Li et al. (2020); Yan et al. (2021); Li et al. (2023b;a). Although voxel-based scene representation has made significant progress in LIDAR-based scene perception, its application in vision-based scene understanding has remained relatively unexplored. MonoScene Cao & de Charette (2022) is one pioneering work to reconstruct 3D scene with only RGB inputs, which projects image features to all possible positions in the 3D space along optical rays, initially obtaining a voxel representation and processing it with a 3D Unet afterward. TPV-Former Huang et al. (2023) further extends it to multi-camera 3D occupancy prediction through a tri-perspective view representation, which lifts and projects image features to three perpendicular planes. However, voxel-based scene representation methods still suffer from high computation complexity due to the large amount of voxels, which limits their application to larger scenes.

### 5.2 BEV-BASED SCENE REPRESENTATION

In recognition of the fact that the height dimension entails less information compared to the other two dimensions, BEV-based scene representation methods implicitly encapsulate height information within each BEV grid to form more compact and efficient scene representations Lang et al. (2019). Recent studies in BEV-based scene representation have focused on refining BEV representations with reliable depth estimation, which can be divided into two main streams. One stream of works adopts BEV queries to implicitly integrate depth information from image features Jiang et al. (2023); Li et al. (2022). Another stream of works explicitly generates a depth map for each input image, and then projects 2D features into 3D space followed by BEV pooling operations Philion & Fidler (2020); Huang et al. (2021); Reading et al. (2021); Liang et al. (2022); Zhang et al. (2022); Li et al. (2023d); Liu et al. (2023). Among them, the pioneering and fundamental work is the Lift-Splat-Shot (LSS) Philion & Fidler (2020) paradigm, which proposes an end-to-end pipeline to "lift" each image individually into a frustum of features, "splat" all frustums into a rasterized BEV grid, and then "shoot" template trajectories into a BEV cost map. Inspired by the LSS paradigm, BEVDet Huang et al. (2021) proposes a general BEV-based pipeline for scene understanding, which consists of four parts: Image-view Encoder, View Transfromer, BEV Encoder, and Task-specific Head. Efforts have been made upon view transformation to obtain better BEV features with precise depth estimation. BEVDepth Li et al. (2023d) introduces a camera-aware depth estimation module together with a depth refinement module to facilitate more accurate depth learning. BEVStereo Li et al. (2023c) further enhances depth estimation with dynamic temporal stereo information, tackling ill-posed issues and improving computational efficiency as well.

## 6 CONCLUSION AND DISCUSSION

We have identified the challenges of sparsity and ambiguity rooted in voxel labels for the 3D occupancy prediction task, which limits the view transformation accuracy and occupancy prediction performance. To address these challenges, this paper introduces the Bi-directional Circulated 3D Occupancy Prediction (BiC-Occ) framework, consisting of two key modules to alleviate the sparsity and ambiguity of voxel labels respectively. The Bi-directional View Transformer module is proposed to approximate a reversible view transformation, alleviating the sparse supervision with self-consistency between 2D image features and 3D BEV representations. In addition, the Circulated

Interpolation Predictor module exploits local geometric structures to align multi-scale BEV representations in a circulated manner, correcting local ambiguity for more accurate 3D occupancy prediction results. These modules together mitigate the sparsity and ambiguity challenges and achieve state-of-the-art performance on the Occ3D-nuScenes Tian et al. (2024) dataset.

**Limitations.** In this work, we have demonstrated that it is possible to compensate for the sparsity and ambiguity of voxel labels with self-consistency regarding 2D-3D representations and multi-scale predictions. We view this as a starting attempt to reduce the dependency on annotated voxel labels, and future work will focus on self-supervised self-consistent occupancy prediction frameworks for efficient and practical applications.

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
