# OpenReview forum: "BiC-Occ: Bi-directional Circulated 3D Occupancy Prediction for Autonomous Driving"
_ICLR.cc/2025/Conference — Submitted to ICLR 2025_

### Official Review · Reviewer_76G2 · 2024-10-15

**Soundness:** 2
**Presentation:** 2
**Contribution:** 2
**Rating:** 3
**Confidence:** 5

**Summary:**

This paper proposes a new method about utilizing the reverse view transformation on the occupancy prediction. The experiments on the nuscenes sound. It seems works and has the best perfomance.

**Strengths:**

1. This paper proposes a Bi-directional Circulated 3D Occupancy Prediction framework, which mainly utilizes the reverse view transformation and SVD idea.
2. The experiment seems surpass the SOTA by a large margin.

**Weaknesses:**

1. Lack of novelty. The reverse projection idea mainly developed from Matrixvt, but this work is not cited in this paper. Besides, there are also some methods such as Inversematrixvt3d or OTOcc have used this idea for occupancy prediction, which also not cited in the paper. Above works also must need to be discussed.
2. Wrong main experiments. Authors should check more about the difference between the nuScene and the Occ-3D nuScenes. the main experiment shows results of the vision-based nuScene lidar segmentation and the Occ-3D nuScenes results, but the title only says it's the
Occ-3D results. Besides, there are a lot of typos in this experiments, like TPVFormer using the R101-DCN. You can check the PanoOcc for the standard benchmark comparison.

Cause of the these reason, I think this paper is coarse, so make the reject decision.

[1] Inversematrixvt3d: An efficient projection matrix-based approach for 3d occupancy prediction
[2] OTOcc: Optimal Transport for Occupancy Prediction
[3] Matrixvt: Efficient multi-camera to bev transformation for 3d perception
[4] PanoOcc: Unified Occupancy Representation for Camera-based 3D Panoptic Segmentation

**Questions:**

Everybody knows the SVD is slow, especially when this paper utilizes it in the large metrices, which time efficiency is concerned a lot. We need the efficiency comparision with the current SOTAs.

---

### Official Review · Reviewer_c5s4 · 2024-11-01

**Soundness:** 3
**Presentation:** 2
**Contribution:** 3
**Rating:** 6
**Confidence:** 3

**Summary:**

BiC-Occ introduces a framework for vision-based 3D occupancy prediction in autonomous driving, enhancing accuracy by addressing label sparsity and ambiguity. Key innovations include a Bi-directional View Transformer for 2D-3D consistency and a Circulated Interpolation Predictor to align multi-scale BEV representations, both contributing to improved voxel prediction and supervision.

**Strengths:**

The Bi-directional View Transformer in BiC-Occ tackles sparsity by constructing reversible view transformation matrices, enhancing self-consistency. This is an interesting idea.

**Weaknesses:**

a) Introducing bidirectional view transformations and multi-scale alignment modules may increase computational complexity. Although the paper emphasizes accuracy improvements, it does not thoroughly discuss potential computational overheads in practical applications.

b) Although the method is novel, the quantitative experiments in the paper do not demonstrate a significant advantage. Further work may require more refined design and parameter tuning to fully showcase the method’s potential.

**Questions:**

a) A detailed analysis of the trade-offs between increased accuracy and computational demands, specifically in resource-constrained environments like edge devices, would be valuable for readers.

---

### Official Review · Reviewer_x6uw · 2024-11-01

**Soundness:** 2
**Presentation:** 2
**Contribution:** 2
**Rating:** 5
**Confidence:** 3

**Summary:**

The paper proposed a Bi-VT solution based on forward\backward projection to solve the sparsity problem in occ labels, and a Circulated Interpolation Predictor to solve the ambiguity problem in occ labels.

**Strengths:**

The experiments on the nuScenes dataset demonstrated the effectiveness of the proposed method.

The sparsity and ambiguity of labels are challenging problems in the field of occupancy, and the proposed method provides a solution with
 Bi-directional View Transformer and Circulated Interpolation Predictor.

**Weaknesses:**

1. About CIRCULATED INTERPOLATION PREDICTOR. I think upsampling and downsampling may increase ambiguity of Occ labels. Due to the "discrete" properties of voxel, upsampling and downsampling will lead to more serious ambiguity at the edge of the object. Can you provide some visualizations or detailed explanations to help the readers understand the motivation of CIRCULATED INTERPOLATION PREDICTOR more clearly.

2. The experiments are only conducted on nuscenes dataset and an additional dataset needs to be added to prove the effectiveness of the method. This experiment can prove whether the proposed method has generalization ability and avoid the impact of data bias.

3. Judging from Table 1, the improvement of the proposed method compared to COTR is relatively small. I think the additional visualizations or experiments should be added to show the advantage of the proposed method over COTR.

4. More visual results are needed. I would like to see how the sparsity and ambiguity of voxel label affect the baseline results respectively, and how the proposed method alleviates the problem. In addition, comparison with other SOTA methods is also necessary.

**Questions:**

see weaknesses above

---

### Official Review · Reviewer_U8pT · 2024-11-03

**Soundness:** 3
**Presentation:** 3
**Contribution:** 2
**Rating:** 5
**Confidence:** 5

**Summary:**

This paper presents the Bi-directional Circulated 3D Occupancy Prediction (BiC-Occ) framework, which consists of two modules to alleviate the sparsity and ambiguity of voxel labels. The Bi-directional View Transformer module is designed to approximate a reversible view transformation. Besides, the Circulated Interpolation Predictor module exploits local geometric structures to align multi-scale BEV representations. The above modules together achieve promising performance on the Occ3D-nuScenes dataset.

**Strengths:**

(1) The paper is clearly written, providing a detailed introduction to the proposed modules and algorithms, making it easier for readers to understand and grasp the key points.

(2) The proposed method also demonstrates good performance on public datasets.

**Weaknesses:**

(1) The bidirectional view-transformation module presented in the paper is not novel, as similar methods have been proposed in FB-BEV/FB-Occ, which diminishes the technical contributions of this work. The authors need to clarify the differences between their work and the aforementioned studies, and further experimental results should be provided to demonstrate the uniqueness of their method.

(2) The experimental contributions of the paper are also relatively weak. The primary experiments are limited to results on the Occ-nuScenes dataset. The ablation study is rather simplistic, lacking unique insights into the hyperparameters of the modules. Additionally, there is no further ablation to clarify the motivation behind the module design.

(3) The paper lacks an analysis of the computational complexity of the proposed method, and the experimental section does not provide information on the number of parameters or inference speed.

**Questions:**

(1) The bidirectional view-transformation module presented in the paper is not novel, as similar methods have been proposed in FB-BEV/FB-Occ, which diminishes the technical contributions of this work. The authors need to clarify the differences between their work and the aforementioned studies, and further experimental results should be provided to demonstrate the uniqueness of their method.

(2) The experimental contributions of the paper are also relatively weak. The primary experiments are limited to results on the Occ-nuScenes dataset. The ablation study is rather simplistic, lacking unique insights into the hyperparameters of the modules. Additionally, there is no further ablation to clarify the motivation behind the module design.

(3) The paper lacks an analysis of the computational complexity of the proposed method, and the experimental section does not provide information on the number of parameters or inference speed.

---

### Official Review · Reviewer_R9AH · 2024-11-11

**Soundness:** 2
**Presentation:** 3
**Contribution:** 2
**Rating:** 3
**Confidence:** 4

**Summary:**

This paper introduces a novel framework named Bidirectional Circulated 3D Occupancy Prediction (BiC-Occ) for addressing 3D occupancy prediction in autonomous driving scenarios. The major contributions of BiC-Occ lie in tackling the sparsity and ambiguity of voxel labels through two key modules:

Bidirectional View Transformer (Bi-VT): This module leverages tensor decomposition and invertibility to handle the sparsity of voxel labels, ensuring self-consistency between 2D and 3D representations.
Circulated Interpolation Predictor (CIP): This module aligns multi-scale voxel representations to address ambiguity across resolutions.

The experiments conducted on the Occ3D-nuScenes validation dataset demonstrate that BiC-Occ outperforms current baseline methods in Scene Completion (SC) and Semantic Scene Completion (SSC) tasks. Ablation studies, parameter analyses, and qualitative analyses further validate the effectiveness of the proposed approach.

**Strengths:**

1.BiC-Occ proposes an innovative approach that combines bidirectional view transformation with circulated interpolation, a novelty in the field of 3D occupancy prediction.
2.Experiments, including ablation studies, parameter analyses, and visualizations, have verified the efficacy of BiC-Occ.
3.The paper is well-structured, providing clear explanations of each module's function.

**Weaknesses:**

1. The paper makes two significant assumptions: (A) F_img and F_BEV can be represented by the Kronecker product of two transition score matrices, and (B) the invertible view transformation can be approximated through decomposition and truncated SVD. It is unclear how these approximations effectively capture complex structural features within a scene, such as rich details or object edges, as well as the nonlinear relationships between 2D and 3D and viewpoint variations. Consequently, I suspect that the improved performance might be largely due to an increase in model size rather than inherent improvements from the proposed method.

2. While the interpolation and 3D convolution operations in the CIP module can align voxel representations across resolutions, these operations may lead to detail loss and edge blurring, particularly when mapping from low to high resolutions. As a result, fine-grained details in high-resolution outputs may not be fully preserved.

3. The parameter analysis in the ablation study focuses only on hyperparameters α and β with a few discrete values, while other potentially impactful parameters (e.g., truncation levels in tensor decomposition or interpolation scales) remain unexplored.

4. Although BiC-Occ claims to address voxel label sparsity and ambiguity, the experiments lack targeted quantitative analyses on these aspects. For example, evaluating the model’s performance specifically in sparse regions or edge-ambiguous areas could more clearly demonstrate BiC-Occ’s effectiveness in addressing these issues.

5. The experiments do not assess BiC-Occ’s computational efficiency or real-time performance. Without such evaluations, it is challenging to determine the method's feasibility for real-world applications, particularly in time-sensitive scenarios such as autonomous driving.

**Questions:**

1. How do the two strong assumptions directly contribute to addressing sparsity and ambiguity? I suspect that the performance gains may primarily result from the increased model size. Could the authors provide additional evidence that demonstrates how the method itself contributes to these improvements?
2. The paper claims to address sparsity and ambiguity issues, but the experiments lack targeted metrics. Could the authors provide results that specifically analyze performance in sparse regions or in areas with ambiguous edges?
3. Have the authors considered evaluating the runtime efficiency of BiC-Occ, particularly for time-sensitive applications such as autonomous driving?
4.. Besides \alpha and \beta,  are there other critical parameters (e.g., truncation levels in tensor decomposition) that might significantly influence model performance? Could the authors provide a more comprehensive exploration of these parameters?

---

### Meta-Review · Area_Chair_wDzw · 2024-12-20

**Metareview:**

This paper presents a Bi-directional Circulated 3D Occupancy Prediction (BiC-Occ) framework to tackle voxel prediction for autonomous driving. Despite achieving reasonable performance, reviewers raise several critical concerns including method assumption (R9AH), paper novelty (76G2, U8pT), experiment completeness (R9AH, 76G2, x6uw), and etc. Also, authors did not provide responses to address these concerns. Therefore, I recommend rejecting this paper.

**Additional Comments On Reviewer Discussion:**

Authors did not attempt to address reviewers' concerns. No response is provided.

---

### Decision · Program_Chairs · 2025-01-22

Reject